# Prostate Cancer’s Silent Partners: Fibroblasts and Their Influence on Glutamine Metabolism Manipulation

**DOI:** 10.3390/ijms25179275

**Published:** 2024-08-27

**Authors:** Pia V. Hönscheid, Gustavo B. Baretton, Martin Puhr, Tiziana Siciliano, Justus S. Israel, Matthias B. Stope, Celina Ebersbach, Alicia-Marie K. Beier, Christian Thomas, Holger H. H. Erb

**Affiliations:** 1Institute of Pathology, University Hospital Carl Gustav Carus, Medical Faculty, Technische Universität Dresden, 01307 Dresden, Germany; pia.hoenscheid@ukdd.de (P.V.H.); gustavo.baretton@uniklinikum-dresden.de (G.B.B.); tiziana.siciliano@uniklinikum-dresden.de (T.S.); 2Core Unit for Molecular Tumor Diagnostics (CMTD), National Center for Tumor Diseases (NCT) Dresden, 01307 Dresden, Germany; 3National Center for Tumor Diseases (NCT), Partner Site Dresden, 01307 Dresden, Germany; christian.thomas@uniklinikum-dresden.de; 4Tumor and Normal Tissue Bank of the University Cancer Center (UCC), University Hospital Carl Gustav Carus, Medical Faculty, TU Dresden, 01307 Dresden, Germany; 5Department of Urology, Medical University of Innsbruck, 6020 Innsbruck, Austria; martin.puhr@i-med.ac.at; 6Department of Urology, University Hospital Carl Gustav Carus, Faculty of Medicine, TU Dresden, 01307 Dresden, Germany; justussimon.israel@uniklinikum-dresden.de (J.S.I.); celina.ebersbach@uniklinikum-dresden.de (C.E.); aliciamarie.beier@uniklinikum-dresden.de (A.-M.K.B.); 7Department of Gynecology and Gynecological Oncology, University Hospital Bonn, 53127 Bonn, Germany; matthias.stope@ukbonn.de; 8UroFors Consortium (Natural Scientists in Urological Research), German Society of Urology, 14163 Berlin, Germany

**Keywords:** tumour microenvironment, cancer-associated fibroblasts, hormone-sensitive prostate cancer, castration-resistant prostate cancer, PCa

## Abstract

Cancer-associated fibroblast (CAF)s in the tumour microenvironment (TME) modulate the extracellular matrix, interact with cancer cells, and facilitate communication with infiltrating leukocytes, significantly contributing to cancer progression and therapeutic response. In prostate cancer (PCa), CAFs promote malignancy through metabolic rewiring, cancer stem cell regulation, and therapy resistance. Pre-clinical studies indicate that targeting amino acid metabolism, particularly glutamine (Gln) metabolism, reduces cancer proliferation and stemness. However, most studies lack the context of CAF–cancer interaction, focusing on monocultures. This study assesses the influence of CAFs on PCa growth by manipulating Gln metabolism using colour-labelled PCa cell lines (red) and fibroblast (green) in a co-culture system to evaluate CAFs’ effects on PCa cell proliferation and clonogenic potential. CAFs increased the proliferation of hormone-sensitive LNCaP cells, whereas the castration-resistant C4-2 cells were unaffected. However, clonogenic growth increased in both cell lines. Gln deprivation and GLS1 inhibition experiments revealed that the increased growth rate of LNCAP cells was associated with increased dependence on Gln, which was confirmed by proteomic analyses. Tissue analysis of PCa patients revealed elevated GLS1 levels in both the PCa epithelium and stroma, suggesting that GLS1 is a therapeutic target. Moreover, the median overall survival analysis of GLS1 expression in the PCa epithelium and stroma identified a “high-risk” patient group that may benefit from GLS1-targeted therapies. Therefore, GLS1 targeting appears promising in castration-resistant PCa patients with high GLS1 epithelium and low GLS1 stromal expression.

## 1. Introduction

Cancer-associated fibroblast (CAF)s are non-neoplastic fibroblasts with pro-tumorigenic properties [1,2]. CAFs generate reactive stroma during carcinogenesis, where pro-tumorigenic alterations in the tumour microenvironment (TME) occur [3,4]. Tumour proliferation is promoted by bidirectional signalling between tumour cells and other cellular entities, mediated by CAF-derived chemokines, cytokines, growth factors, and exosomes within the TME. This interaction also triggers immune evasion of cancer cells [5,6]. Furthermore, CAFs participate in cancer progression by modulating the metabolic function of cancer cells. The metabolic dialogue between cancer cells and CAFs occurs mainly via soluble factors, extracellular matrix, and direct contact [7,8]. Breast cancer studies revealed that CAFs fuel cancer cells with organic substances such as pyruvate and amino acids (e.g., glutamine), which fill the citric acid cycle and subsequent mitochondrial respiration [7,9]. Glutamine (Gln) is converted into glutamate and enters the citric acid cycle, fuelling biosynthetic pathways that produce lipids, proteins, and nucleic acids, thus promoting cancer progression [9,10,11,12]. In breast cancer, it could be shown that Gln deprivation promotes CAF migration and invasion, facilitating tumour epithelial cell migration toward nutrient-rich areas [13]. In lymphoma, CAFs fuel cancer cells with amino acids (e.g., glutamine) and with organics such as pyruvate, which is at the crossroads between glycolysis and mitochondrial oxidative phosphorylation [14]. By engaging in bidirectional signalling with epithelial tumour cells and other cells mediated by CAF-derived cytokines, chemokines, growth factors, and exosomes within the TME, CAFs facilitate tumour proliferation and induce immune evasion of cancer cells in gastrointestinal cancer [5]. In pancreatic ductal adenocarcinoma, cancer cells can proliferate in nutrient-deprived conditions when supplied with Gln by extracellular vesicles originating from CAFs [15].

CAFs are the predominant cell type within the prostate cancer (PCa) stroma [2,16]. PCa is a major clinical problem, as it has the second-highest incidence in the male population worldwide and is the fifth leading cause of cancer-related deaths [17]. PCa treatment depends on the tumour stage, patient age, general health status, and tumour risk assessment. The treatment options recommended by the guidelines include active surveillance, surgery, and radiotherapy for localised PCa with a curative intent [18]. However, only palliative hormone therapy and chemotherapy are available for advanced and non-localised PCa [19]. Despite an initially favourable response to established therapies, treatment failure eventually occurs, leading to the progression of hormone-sensitive PCa (HSPC) to rapidly growing and aggressive castration-resistant PCa (CRPC). Therefore, translational and basic research has focused on identifying novel target structures in PCa and developing novel therapeutic strategies.

In PCa, CAFs promote malignancy through metabolic rewiring [7]. Specifically, CAF-PCa crosstalk enhances mitochondrial transfer and induces oxidative phosphorylation (OXPHOS) addiction in PCa cells. Metabolic reprogramming is essential for PCa progression, tumour growth maintenance, and therapy resistance [20,21]. Recent studies have shown that targeting amino acid metabolism in cancer cells reduces cancer cell proliferation and stemness [22]. Mainly targeting Gln metabolism has shown promising results in in vitro and in vivo studies [11,20,23,24]. However, the influence of this inhibitor in a CAF-PCa cell co-culture system has not yet been investigated.

Glutaminase (GLS) plays an essential role in the Gln metabolism. GLS catalyses the transformation of Gln into glutamate, which is subsequently converted to α-ketoglutarate, a component of the TCA [25]. Two primary glutaminase types have been identified: glutaminase 1 (GLS1, also known as kidney-type) and glutaminase 2 (GLS2, also known as liver-type). GLS1 is mainly expressed in prostate cancer (PCa), where its levels increase with the tumour stage and progression, as reported by Myint et al. [26]. In contrast, GLS1 and GLS2 appear to serve opposing roles in tumour development, attributed to various regulatory mechanisms and their distinct immunological, kinetic, and molecular properties [27]. Different regulatory mechanisms under various conditions influence the expression of these enzymes. For instance, the transcription factor Myc enhances GLS1 expression, promoting the proliferation of tumour cells in PC3 PCa cells [28]. The specific mechanisms through which c-Myc regulates glutaminolysis, particularly the upregulation of GLS1, are not fully understood. However, in PC3 cells, it has been shown that c-Myc elevates GLS1 expression through post-transcriptional modifications mediated by microRNAs, specifically miR-23a and miR-23b.

The interaction between Myc and GLS2 is subject to ongoing debate, varying with cell type and tumour characteristics. For example, GLS2 is downregulated in renal cell carcinoma induced by Myc, whereas it is upregulated in T-lymphocytes by c-Myc [29,30]. GLS2, which plays a role in tumour metabolism and antioxidant defence in both stressed and non-stressed conditions, is primarily controlled by the transcription factor p53 (Hu et al., 2010). Additional transcription factors such as p73 and p63 also regulate GLS2 expression during processes like neuronal differentiation, epidermal differentiation, and tumorigenesis under conditions of high oxidative stress [31,32]. Apart from regulation by c-Myc or p53, specific metabolic products also influence the expression of these isoenzymes. For example, GLS1 activity is increased by higher phosphate levels and decreased by glutamate. In contrast, GLS2 activity increases with low phosphate levels and is unaffected by glutamate [33].

Therefore, the effects of Gln deprivation on the stemness and proliferation of PCa cells were examined. Additionally, the project aimed to visualise the spatial distribution of CAF metabolism within PCa tissue sections to improve pathological assessment based on Gln metabolism. To this end, a two-colour PCa cell (red) and CAF (green) co-culture system was established. This model helped assess the influence of fibroblasts on PCa proliferation and clonogenic potential. Furthermore, the role of CAFs in GLS1 pathways associated with proliferation and clonogenic potential was investigated by manipulating glutamine (Gln) metabolism using Gln deprivation and the GLS1 inhibitor CB-839.

## 2. Results

### 2.1. Established Prostate Cancer Cell Lines Expressing mKATE2-NLS to Trace Cell Proliferation and Clonogenic Potential

To distinguish between fibroblasts and PCa cells in a co-culture system using an S3 Incucyte Live-Cell Analysis System (Sartorius AG, Göttingen, Germany), the PCa cell lines LNCaP and C4-2 were transduced with mKATE2 fluorescent protein (excitation maximum of 588 nm, emission maximum of 633 nm) linked to a nuclear localisation signal (NLS). LNCaP and C4-2 cells were transduced with multiplicities of infection (MOI) of 2.5, 5, and 10. mKATE2-NLS positivity was assessed using a BZ-X800 microscope (Keyence GmbH, Neu-Isenburg, Germany) and the BZ-X800 Analyzer software 1.3.0 (Keyence GmbH, Figure 1A,B). LNCaP cells showed the highest mKATE2-NLS positivity (79.1% ± 0.1) at an MOI of 2.5 (Figure 1A,B). C4-2 cells showed the highest mKATE2-NLS positivity (100%) at an MOI of 5 (Figure 1A,B). Cells with the highest mKATE2-NLS positivity were selected for 10 µM blasticidin treatment. To validate positive blasticidin selection, mKATE2-NLS positivity was assessed again after the selection process, resulting in a significant increase in LNCaP up to 87.2% ± 0.03 (Figure 1C), whereas C4-2 stayed at 100% mKATE2-NLS positivity (Figure 1C). To validate whether mKATE2-NLS nuclear expression during proliferation correlated with the increase in confluence of unlabelled cells, transduced cells were cultured in triplicate in 96-well plates, and cell confluence and mKATE2-NLS numbers were determined for 5 days using the S3 Incucyte Live-Cell Analysis System (Sartorius AG, Figure 1D). All the cell lines showed a significantly strong correlation (LNCaP r = 0.97, C4-2 r = 0.99) between the observed variables. Therefore, the mKATE2-NLS transduced cell lines were deemed suitable for co-culture experiments and were utilised in all further experiments. 

### 2.2. Cancer-Associated Fibroblasts Sensitise the Hormone-Sensitive LNCaPcell Line to Glutamine Deprivation

To assess the impact of CAFs on Gln deprivation in LNCaP and C4-2 cells, the cell lines were grown in the presence and absence of CAFs and Gln (0 mM and 2 mM) for 96 h. Changes in PCa cell proliferation were determined by mKATE2-NLS positive nuclei counting using the S3 Incucyte Live-Cell Analysis System (Sartorius AG, Göttingen, Germany), and changes in proliferation rates were displayed as changes in AUC (Figure 2A,B and Appendix A), as suggested by Duan et al. [34]. Co-culturing LNCaP cells with CAF cells significantly increased the proliferation of LNCaP cells (Appendix A), whereas the C4-2 cell’s proliferation (Appendix A) was not affected by CAFs (Appendix A). Without CAFs, Gln deprivation had a negligible effect on LNCaP proliferation (Figure 2A). In the presence of CAFs, Gln deprivation significantly reduced cell proliferation. However, the growth rate of C4-2 cells was significantly reduced regardless of the presence or absence of CAFs (Figure 2B). In addition to proliferation assays, clonogenic assays were performed to study the influence of CAFs on Gln deprivation on the ability of a single LNCaP or C4-2 cell to grow into a colony (Figure 2C,D). The analysis revealed that CAFs increased the CFE of both cell lines (Appendix A). Moreover, the CFE-reducing effects of the Gln deprivation on LNCaP (Figure 2C) and C4-2 cells (Figure 2D) were diminished by CAF cells.

### 2.3. Cancer-Associated Fibroblasts Reprogram LNCaP Cell’s Metabolic Needs

To investigate how CAFs can sensitise hormone-sensitive LNCaP cells to Gln deprivation, the effect of CAFs on the LNCaP proteome was assessed. To this end, eGFP2-nls transduced CAF cells were co-cultured for 96 h with mKATE2-nls transduced PCa cells and separated by FACS. The proteomes of the separated cells were analysed by mass spectrometry and compared with those of the corresponding transduced monoculture cells (Figure 3A). To investigate the differential gene expression induced by CAF cells after 96 h of treatment, volcano plots (Figure 3B and Appendix A) comparing the co-cultured cells with the corresponding monocultured cells were used to illustrate the relationship between fold-change and statistical significance (adjusted *p*-value) for all analysed proteins. Co-culturing resulted in a significant change (adjusted *p*-value <0.05) in 738 proteins and 182 proteins in LNCaP and CAF cells, respectively (Figure 3B and Appendix A). To elucidate the biological pathways associated with the differentially expressed genes (DEGs), PathfindR analysis for pathway enrichment analysis was performed using an active subnetwork-oriented approach to identify significantly enriched pathways (Figure 3C and Appendix A). The analysis identified several key pathways significantly enriched in our dataset, including metabolic pathways with highly significant gene enrichment in LNCaP co-cultured cells and several pathways influencing the extracellular matrix in CAF-co-cultured cells (Appendix A). The gene set enrichment analysis (GSEA) analysis of different pathways of the metabolic group of the Reactome database revealed that the gene set/pathway “metabolism of amino acids and derivatives” was significantly enriched in co-cultured LNCaP, with a normalised enrichment score (NES) of 1.72 (Figure 3D,E).

### 2.4. Cancer-Associated Fibroblasts Reduce the Inhibitory Effect of CB839 on Clonogenic Potential

Gln is a non-essential amino acid that makes its removal from the body impossible. Several inhibitors have been developed to target Gln metabolism, of which the selective GLS1 inhibitor CB839 is the most promising [12,35,36,37]. Therefore, the influence of CAFs on the inhibitory effects of CB-839 cells on LNCaP and C4-2 cell proliferation and clonogenic potential was examined (Figure 4). To ensure that both cell lines expressed the therapeutic target structure of CB839, the expression of GLS1 in LNCaP and C4-2 cells was analysed using qPCR and Western blotting. Both cell lines showed comparable expression levels of GLS mRNA (Figure 4A) and GLS1 protein (Figure 4B,C). To assess the influence of CAFs on the effects of CB-839, cell lines were cultured in the presence or absence of CAFs and CB-839 for 96 h (Figure 4D,E). As shown previously, CB-839 treatment led to a minor reduction in LNCaP cell proliferation without CAFs.

In contrast, CAFs increased the inhibitory effect on cell proliferation (Figure 4D). This result is also reflected in dose–response experiments after 96 h (Appendix A). Consistent with the Gln deprivation results, CB839 reduced proliferation to the same extent in C4-2 cells alone and co-cultured with CAFs (Figure 4E). As CAFs could rescue the effects of Gln deprivation on clonogenic potential, the effects of CAFs on the clonogenic potential of CB839 were also assessed (Figure 4F,G). In line with the results shown for Gln deprivation (Figure 4H,I), the presence of CAFs reduced the inhibitory effects on the clonogenic potential of CB839. 

### 2.5. Low GLS1 Expression in Stroma and High Expression in Epithelial Cells Is a Marker for Bad Prognosis

Previous studies have indicated the role of GLS1 as a marker for therapy resistance and poor prognosis [11,20,23]. However, these studies did not include GLS1 expression in stroma. Therefore, previously published GLS1 staining of the Dresden cohort was re-evaluated using GLS1 staining of the stroma and epithelial cells [11]. GLS1 expression was elevated in cancerous epithelial cells (Figure 5A). 

Additionally, the assessment of GLS1 staining in stromal cells revealed elevated GLS1 expression in cancerous stroma areas (Figure 5B). Correlation analysis (Figure 5C) demonstrated a significant moderate correlation (r = 0.67; *p* < 0.01) between GLS1 expression in benign stroma and epithelium and a significant moderate correlation (r = 0.75; *p* < 0.01) between GLS1 expression in cancer stroma and epithelium. In line with previous results, Kaplan–Meier analysis revealed a median overall survival (MS) of 64 months for patients with low GLS1 and 45 months for those with high GLS1 expression in the prostate epithelium (Figure 6A). For prostate stroma, Kaplan–Meier analysis revealed a MS of 64 months for low GLS1 expression and 30 months for high GLS1 expression (Figure 6B). To investigate whether a combination of GLS1 epithelium and stoma combination would have any predictive value, four groups (high/high, high/low, low/high, and low/low) were created based on GLS1 expression in the prostate epithelium and stroma (Figure 6C). Group analysis revealed an MS for low/low at 64 months, low/high at 57 months, high/low at 30 months, and high/high at 100 months. As the high/low group showed the lowest MS, it was categorised as a high-risk group. Subsequently, Kaplan–Meier analysis of the high-risk group compared with the rest of the cohort (low-risk) revealed a significant reduction in MS to 30 months (high-risk) from 65 (low-risk) months (hazard ratio: 0.294) (Figure 6D).

## 3. Discussion

Several efficient treatment options with curative intent are available for localised PCa; however, approximately 20–40% of patients experience tumour recurrence, and therapy resistance develops [38,39,40,41,42]. Therefore, research strategies are needed to detect cancer progression and hormonal resistance to reveal possible target windows to inhibit tumour spreading. 

In PCa, metabolic changes enable cancer cells to utilise unconventional nutrients to meet their energy needs and sustain proliferation [12,43]. Increased amino acid metabolism, such as Gln, supports redox balance, energy regulation, biosynthesis, and homeostasis in PCa, making it a promising therapeutic target. Targeting Gln metabolism has shown promising results [11,12,20,44,45]. CAFs within the TME play a crucial role in tumour growth by supplying nutrients, such as amino acids, to cancer cells [7,46]. Therefore, this study aimed to investigate the effects of amino acid metabolism inhibitors of GLS1 on CAFs and their interaction and impact on the stemness and proliferation of PCa cell lines.

The TME influences PCa progression and therapeutic response, with CAFs exhibiting pro-tumourigenic properties [47]. Co-culture experiments showed that CAFs enhance the proliferation of hormone-sensitive LNCaP cells. This finding aligns with previous studies by Yu et al., Sun et al., and Tagat et al., all of whom reported an increased LNCaP proliferation and viability in the presence of CAFs [2,48,49].

Conversely, CAFs had little effect on the proliferation of CRPC cell line C4-2, unlike previous studies showing significant enhancement by CAFs [49,50,51]. These studies used conditioned supernatants or transwell systems, missing direct cell–cell interactions, which may reduce growth due to contact inhibition and increase cell cycle inhibitors, such as p27 and p21 [52,53,54,55]. Here, fibroblasts increased only LNCaP cell proliferation but not the CRPC cell line C4-2, suggesting that fibroblasts play a proliferative role in HSPC. In CRPC, CAFs may maintain and protect against cancer progression, potentially forming niches for therapy resistance and correlating with unfavourable clinical outcomes [56,57].

This hypothesis was supported by clonogenic assays, which revealed that CAFs increase CFE in both hormone-sensitive and castration-resistant PCa cell lines. This result indicates that CAFs enhance stemness and regeneration, contributing to tumour progression [58]. Similar results have been observed in breast cancer cell lines co-cultured with serum-activated fibroblasts, suggesting that these cells promote clonogenic growth, therapy resistance, and metastasis [59,60].

CAFs have been reported to promote PCa progression via metabolic rewiring, responsible for tumour growth, progression, maintenance, and therapy resistance [7,28]. Proteomic and GSEA analyses performed in this study confirmed the influence of metabolic pathways on the increase of hormone-sensitive cell line LNCaP dependency on amino acid metabolism, particularly Gln. This increased metabolic requirement may be coupled with the increased proliferation of LNCaP cells, resulting in a higher energy requirement. In turn, the presence of LNCaP cells mainly influenced signalling pathways in CAFs that affect the extracellular matrix (ECM). CAFs have been shown to alter the expression of ECM and adhesion-related proteins, thereby influencing ECM remodelling [61]. Moreover, collagen-related alterations in tissue were linked to clinically significant prostate cancer at primary diagnosis, indicating that the influence of cancer cells on CAFs may be responsible for the reported ECM changes, which could serve as a potential biomarker [62]. Several factors may be responsible for this reprogramming, including extracellular vesicles, cytokines, and growth factors that are secreted from CAFs. Cytokines like IL-6 have been reported to influence metabolic processes in cancer cells and alter LNCaPs response to other soluble factors [63,64,65]. However, also direct cell–cell contact may induce metabolic rewiring in the LNCaP cells.

To meet their elevated energy demands, cancer cells undergo metabolic reprogramming, resulting in a phenomenon known as glutamine addiction [66]. In vivo and ex vivo experiments conducted on cellular models of non-small cell lung cancer, glioblastoma, and hepatocellular carcinoma have demonstrated a pronounced dependency of these cells on Gln [67]. In the context of prostate cancer (PCa), specifically the castration-resistant prostate cancer (CRPC) phenotype, such dependency appears to be limited to aggressive forms [12,24]. This observation suggests a metabolic adaptation during the acquisition of resistance to androgen receptor pathway inhibitors. The present study corroborates these findings, illustrating an enhanced glutamine sensitivity in the CRPC-specific LNCaP sub-cell line C4-2. Furthermore, variability in metabolic phenotypes has been observed across different tumours and within sub-types of the same cancer. Such variations have been reported in diverse cancer types, including non-small cell lung cancer, glioblastoma, and breast cancer. This variation underscores the complexity of metabolic reprogramming in cancer and highlights the potential for targeted therapeutic interventions based on specific metabolic dependencies.

Gln has multiple sources in the human body, so it cannot be eliminated. The selective GLS1 inhibitor CB-839 was used to examine its influence on cell proliferation and clonogenic potential. CB-839 has been tested in several clinical trials for advanced solid tumours, including PCa, encouraging clinical activity and tolerability [12,68,69]. In line with the Gln deprivation results, LNCaP cells were more sensitive to treatment in the presence of CAF cells, validating the increased Gln requirement induced by CAFs. Independent of CAFs, the CRPC cell line C4-2 exhibited similar sensitivity to CB-839. This finding is consistent with previous studies reporting that CB-839 inhibits cell proliferation to a greater extent in advanced PCa cells than in hormone-dependent cells [23,70]. Like Gln deprivation, CB-839 treatment reduced the CFE across all investigated cell lines, showing an increased influence on the CFE of CRPC cells compared with the hormone-sensitive cell line LNCaP. This conclusion is supported by the findings of Xie et al., which demonstrated a radiation-resistant phenotype of C4-2 using gene expression analyses that revealed differential regulation of genes involved in cell cycle arrest and DNA repair [71]. Radioresistance has been linked to increased Gln dependency in PCa [20].

Furthermore, C4-2 cells have been characterised by reduced sensitivity to androgen deprivation and antiandrogen treatments, reinforcing their classification as castration-resistant prostate cancer (CRPC) [72]. Both forms of resistance have been associated with an increased dependency on glutamine (GLN) metabolism, which may explain the heightened sensitivity of C4-2 cells to GLN deprivation [24]. These findings on cell proliferation and CFE induced by CB-839 treatment align with previous observations, indicating that Gln plays a more maintenance-oriented role in CRPC [23]. This conclusion can be supported by the work of Xu et al., who showed a GLS1 isoform switch from KGE to GLC in CRPC development, which is associated with more aggressive tumour progression and higher Gln dependency [23].

Because CAFs have been shown to create a niche that promotes the clonogenic potential of all selected PCa cells, the influence of CAFs on Gln deprivation and GLS1 inhibition was assessed. The results revealed that, after inhibition of GLS1 activity, CAFs diminished the Gln-reducing effect in all tested cell lines. This result indicates that GLS1 plays a role in the regenerative and clonogenic potential of PCa, independent of hormone status, and may be an important factor in tumour recurrence and metastasis.

As Gln deprivation and GLS1 inhibition could only be partially counteracted by fibroblasts, their expression was investigated in a large cohort of PCa tissues to assess its value as a therapeutic target. In line with other studies, increased expression of GLS1 has been observed [11,23,26,73]. Correlation analysis of the expression showed a moderate link between the epithelium and stroma, indicating no direct relationship between increased expression in the malignant epithelium and stroma. According to MS, the Kaplan–Meier analysis of GLS1 expression in the epithelium and stroma revealed only a minor advantage for patients with low GLS1 expression. Similar results were reported by Myint et al., who did not show a statistical difference in MS but showed a trend toward worse disease-free survival with high GLS mRNA levels [26]. However, Kaplan–Meier analysis of GLS1 expression in epithelium and stroma combined revealed a “high-risk” patient group with low stroma and high cancer GLS1 expression, which had a statistically shorter MS than the other patients. GLS1 expression is widespread among patients in malignant areas, so these “high-risk” patients may benefit most from therapeutic intervention against GLS1. However, this result needs to be validated in an independent patient cohort.

Moreover, this result reflects the high heterogeneity of PCa, one of the most significant issues in precision therapeutic interventions, highlighting the urgent need for suitable biomarkers for personalised medicine in PCa. However, targeting GLS1 and the Gln metabolism has already been reported to induce a metabolic shift to glycolysis [74]. This shift may be a reason for the mediocre success of Gln inhibitors in clinical trials [12,24,35,69]. Therefore, combination strategies should be considered to develop suitable therapeutic strategies.

This feasibility study developed a two-colour co-culture PCa cell (red) and fibroblast (green) model to investigate the direct influence of CAF and Gln metabolism on the growth behaviour of HSPC and CRPC cell lines. It was demonstrated that CAFs lead to metabolic reprogramming of the HSPC cell line LNCaP, making it more dependent on Gln for its growth behaviour. While providing valuable insights, this study is subject to several limitations that warrant mention. Firstly, although the proteomics analysis indicated various alterations in LNCaP cells and co-cultured cancer-associated fibroblasts (CAFs), the direct influence of CAFs on castration-resistant prostate cancer (CRPC) was not extensively investigated. Future studies should consider a detailed exploration of this interaction better to understand the role of CAFs in CRPC progression. Secondly, comprehensive metabolomic analysis is recommended to provide deeper insights into the metabolic reprogramming induced by CAFs. Such studies would enhance our understanding of the metabolic dynamics within the tumour microenvironment and could unveil potential therapeutic targets.

Additionally, this study has not thoroughly examined the impact of direct cell–cell contacts. Comparative analyses involving co-culture systems and conditioned media could elucidate the specific contributions of soluble factors versus direct cellular interactions to tumour behaviour. The incorporation of 3D culture models, such as spheroids, should also be considered in future research. It has been documented that 3D cultures can significantly alter cancer cell metabolism, thus providing a more physiologically relevant model system that better mimics in vivo conditions [75]. Furthermore, xenograft experiments present challenges, as murine stroma can infiltrate human tumour cells, potentially confounding results. Alternative models that minimise or account for these interspecies interactions need to be developed to enhance translational relevance. 

Despite these limitations, this study successfully demonstrates the feasibility of a co-culture system and underscores the importance of replicating complex tumour architectures in vitro. This approach is crucial for bridging the gap between laboratory findings and clinical applications, thereby facilitating the translation of research from bench to bedside. Moreover, failed clinical trials such as the ENTRATA Trial or the CANTATA randomized clinical trial should be re-evaluated using GLS1 expression in TME and epithelial cells.

## 4. Materials and Methods

### 4.1. Cell Lines

Hormone-sensitive LNCaP, hTERT PF179T CAF (CAF), and 293T cell lines were obtained from the American Type Culture Collection (ATCC). Castration-resistant C4-2 cells were kindly provided by Prof. Thalmann (University of Bern, Bern, Switzerland) [76]. Mycoplasma testing was routinely performed using the Mycoalert detection assay (Lonza, Basel, Switzerland) and cell line authentication was performed annually using STR profiling. 

### 4.2. Lentivirus Production, eGFP-NLS and mKATE2-NLS Nuclei Labelling, and Image Cytometry

For lentiviral production, 293T cells were transfected with psPAX2, pVSV-G, and pLenti6.4-EF1a-mKATE2-NLS or pLenti6.4-EF1a-eGFP-NLS expression vectors at a ratio of 3:1:4 using ViaFect™ transfection reagent (Promega GmbH, Walldorf, Germany) according to the manufacturer’s instructions, as described by Beier et al. 2024 [11]. For cell line transduction with mKATE2-NLS, target cell lines were incubated overnight (ON) with lentivirus and a multiplicity of infection of 2.5, 5, and 10. After one week, mKATE2-NLS positivity was determined by immunofluorescence (Figure 1A,B). All the cells were counterstained with 1 μM Hoechst 33,342 (Thermo Fisher Scientific GmbH, Frankfurt, Germany). mKATE2-NLS (excitation peak 588 nm and emission peak 633 nm) and Hoechst 33,342 (excitation peak 352 nm and emission peak 454 nm) positive cells were detected using a KEYENCE BZ-X800 microscope (Keyence GmbH, Neu-Isenburg, Germany) and analysed using BZ-X800 Analyzer software (Keyence GmbH, Neu-Isenburg, Germany). The cell lines with the highest positivity were subsequently used for blasticidin selection (Figure 1C).

### 4.3. Gln Deprivation and CB-839 Treatment

Cells were seeded in their growth media (2 mM Gln) for 24 h for Gln deprivation proliferation experiments. Subsequently, media was discarded, and cells were cultured in RPMI1640 (Cat# R0883, Sigma Aldrich, Merck KGaA, Darmstadt, Germany) supplemented with 10% FBS with or without 2 mM L-glutamine (Gln, Cat# G8540, Sigma Aldrich) for 96 h or 120 h. Changes in PCa cell proliferation were assessed by red nuclei counting using the S3 Incucyte^®^ Live-Cell Analysis System. For CB-839 (Cat# S7655, Selleck Chemicals) experiments, cells were seeded in RPMI1640 containing 2 mM Gln for 24 h. Subsequently, cells were treated with different concentrations of CB-839 solved in DMSO 96 h. The CB-839 untreated controls were treated with the appropriate amount of DMSO to exclude solvent effects. 

### 4.4. Proliferation Assay with the IncuCyte^®^ S3 Live-Cell Analysis System

Cell proliferation was measured by mKATE2-NLS labelled nuclei counting determination using the IncuCyte S3 Live-Cell Imaging System (Sartorius AG, Goettingen, Germany). The cells were seeded in 96-well clear flat-bottom plates (Corning GmbH, Kaiserslautern, Germany) and incubated ON at 37 °C and 5% CO_2_. Subsequently, the plates were treated and placed into the IncuCyte S3 Live-Cell Imaging System live imaging system and scanned every 6 h for 5 consecutive days. Confluence and cell number were analysed using IncuCyte 2023C analysis software by measuring the growth area or counting the mKATE2-NLS labelled nuclei. Cell proliferation was expressed as increased cell confluence or number compared with the first scan time point.

### 4.5. Clonogenic Assay

The clonogenic assay was performed as described by Franken et al. [77]. First, CAFs were seeded in 6-well plates and allowed to adhere for 24 h. Subsequently, cancer cells were either added to the previously seeded CAFs or individually in 6-well plates. Concurrently, the cells were treated with 2 mM Gln (control group), 0 mM Gln, or 2 mM Gln + 1 µM glutaminase inhibitor (CD-839) and incubated for 10 days. The cell medium was changed after 4 days of incubation. Colonies were fixed with 3.7% formaldehyde solution (Merck KGaA, Darmstadt, Germany) at room temperature (RT) for 15 min, washed multiple times with 2 mL PBS, and covered with 1 mL PBS. Colony amounts were visualised and determined using a compact fluorescence microscope BZ-X800E (Keyence GmbH, Neu-Isenburg, Germany). Colony-forming efficiency was evaluated using BZ-X800 Analyzer Software (Keyence GmbH, Neu-Isenburg, Germany). Cell colonies were defined as those containing at least 50 cells per colony. The minimum size required was set individually for each cell line. Colony-forming efficiency (CFE) and survival fraction were calculated as described by Franken et al. [77]. 

### 4.6. Western Blot

Adherent cells were harvested with RIPA buffer complete protease inhibitor cocktail (Roche Applied Science, Penzberg, Germany), lysed, and protein concentrations were determined as previously described [78,79]. As previously described, 20 µg of protein was used for the Western blot analysis [78,79]. Glutaminase-1/GLS1 (E9H6H) XPR rabbit mAb (1:5000, LOT: 19/05-G4cc-C5cc, Cat#: NB600-502, Cell Signaling Technology, Frankfurt am Main, Germany), mouse monoclonal anti-GAPDH (6C5cc) (1:10,000, LOT: 1, Cat#: 88964, Bio-Techne GmbH, Wiesbaden, Germany), IRDye 680RD goat anti-mouse (1:20,000, LOT: D30207-05, Cat#: 926-68070, Li-COR Biosciences GmbH, Bad Homburg vor der Höhe, Germany), and IRDye 800 CW goat anti-rabbit (1:20,000, LOT: D30307-15, Cat#: 926-32211, Li-COR Biosciences GmbH) were used for protein detection. Signal visualisation was performed using an Odyssey M system (Li-COR Biosciences GmbH) and analysed using Image Studio 6.0 software (Li-COR Biosciences GmbH). Uncropped Western blot images are displayed in the Appendix A. Raw images files are displayed in Appendix A.

### 4.7. Data Preparation, Imputation, Overrepresentation Analysis (ORA), and Gene Set Enrichment Analysis (GSEA) of the Mass Spectrometry Data

The cells were lysed using RIPA buffer for mass spectrometry, and the protein concentration was determined as described previously [78,79]. Lysis buffer (10% SDS, 100 mM TEAB, pH 8.5) was then added to the sample at a ratio of 1:1. The core facility “Mass Spectrometry and Proteomics TU Dresden” processed and normalised the samples as previously described [80,81]. Further analysis were performed using R and R Studio [82,83]. The gene names were separated and filtered for missing values, and duplicate genes were excluded using tidyr und dplyr packages [84]. Imputation was performed using the missForest package [85,86]. Imputed data were processed using the limma package, which uses moderated t-statistics and Benjamin Hochberg multiple analysis correction. Volcano plots were computed to explore the data, visualising significantly up- or downregulated genes with a *p*-value of <0.05 [87,88]. Differentially expressed genes with an adjusted *p*-value of <0.05 were processed using the pathfindR package with the Reactome gene set [89]. This package utilises a protein–protein interaction network (PIN) and performs a one-sided hypergeometric test on the active subnetworks. For gene set enrichment analysis (GSEA), all differentially expressed genes of LNCaP monoculture vs. co-culture were ranked according to sign (logFC) × log10 (adjusted *p*-value) and sorted in descending order. Gene symbols were converted to Enrez IDs using clusterProfiler’s bitr function. This ranked list was processed by ReactomePA using the function gsePathway with the parameter by =‘fgsea’ [90,91,92,93]. Afterwards, bar plots and GSEA plots were created. Pathways with an adjusted *p*-value < 0.05 were considered significant. The R script, session info, and packages used were deposited into GuitHub at https://doi.org/10.5281/zenodo.13208429.

### 4.8. Patients and Study Design 

This cohort contained 108 tissue specimens from PCa patients undergoing palliative TURP (Table 1) [11,94,95]. Matched benign samples were excised from the histologically confirmed non-malignant regions of 76 patients. This study was approved by the local institutional review board of the Faculty of Medicine of Technische Universität Dresden (ethics vote EK43022017). This study followed the Declaration of Helsinki and the ICH Harmonized Tripartite Guideline for Good Clinical Practice.

### 4.9. Immunohistochemistry

Immunohistochemistry (IHC) was performed as described previously [11]. GLS1 IHC was performed using the Ventana BenchMark device (Roche, Vienna, Austria). The following antibodies were used: GLS1 (E9H6H) RabMab XP^®^ (1:800; LOT: 19/05-G4cc-C5cc, Cat#: NB600-502; Cell Signaling Technology, Frankfurt am Main, Germany). As previously described, the evaluation was performed using the modified “quick-score” protocol [11]. 

### 4.10. qPCR

mRNA and cDNA were isolated as previously described [96,97]. Quantitative PCR (qPCR) was performed using GoTAq Probe qPCR master mix (Promega GmbH, Mannheim, Germany). To this end, the following qPCR-mix was used for each sample: 2.5 µL RNAse and DNase-free distilled water, 0.5 µL 20× TaqMan assay (Thermo Fisher Scientific GmbH, Frankfurt, Germany), and 5 µL Go Taq Probe qPCR master mix. cDNA (2 µL) was added to 8 µL qPCR mix, pipetted into a 96-well microtiter plate, and placed into a LightCycler 480 (Roche, Mannheim, Germany). qPCR was performed for 45 cycles using the recommended run template (denaturation step at 95 °C for 10 min, amplification step at 95 °C for 15 s, and subsequently at 60 °C for 60 s, and a final cooling step at 40 °C for 1 min). The data were analysed using the ΔΔCt method. Data were expressed as 2^−ΔCt(Gene of Interest—HPRT1)^. The following TaqMan assays (all Thermo Fisher Scientific GmbH, Frankfurt, Germany) were used for qPCR: *GLS* (Hs01014020_m1), and *HPRT1* (Hs02800695_m1).

### 4.11. Statistical Analysis

Prism 10.2.3 (GraphPad Software, San Diego, CA, USA) was used for statistical analyses. Data were presented as mean ± SD or mean ± SEM to estimate the mean in repeated experiments. The area under the curve (AUC) was calculated to compare growth rates, as previously described [11,34,98]. The Kolmogorov–Smirnov and D’Agostino–Pearson omnibus normality tests determined the Gaussian distribution. Student’s *t*-test (two-sided) and one-way analysis of variance (ANOVA) with Šídák correction were used to identify significant differences. Unless otherwise noted, all experiments were performed with at least three biological replicates. Statistical significance was set at *p* ≤ 0.05, and statistical significance was indicated by asterisks (* *p* ≤ 0.05, ** *p* ≤ 0.01, *** *p* ≤ 0.001).

## Figures and Tables

**Figure 1 ijms-25-09275-f001:**
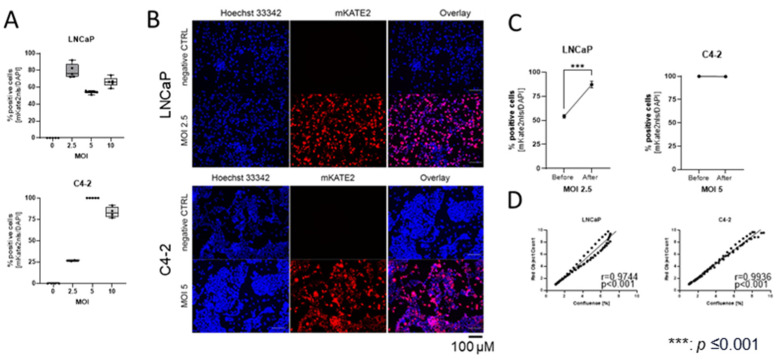
Establishment of prostate cancer (PCa) cell lines expressing mKATE2-NLS. (**A**) LNCaP and C4-2 cells were transduced with a MOI of 2.5, 5, and 10. Positive cells were detected using the Keyence microscope and analysed with the BZ-X800 Analyzer software. The graphical illustration of the mKATE2-NLS positive cells for each cell line was plotted as a box and whisker plot (min to max) of the six technical replicates. (**B**) Representative pictures of each cell line before blasticidin selection are shown of the chosen MOI compared with the untransduced cells (negative CTRL). The scalebar represents 100 μm. (**C**) Analysis of positive cells before and after blasticidin selection. mKATE2-NLS positive cells were detected using the BZ-X800 microscope (Keyence GmBH, Neu-Isenburg, Germany) and analysed with the BZ-X800 Analyzer software (Keyence GmbH). The graphical illustration of the mKATE2-NLS positive cells for each cell line was plotted as the mean ± SD of six technical replicates. An unpaired student’s T-test was used to detect significant differences. *p*-values ≤ 0.05 were considered significant. ***: *p* ≤ 0.001. (**D**) Correlation analysis of cell number determined by mKATE2 counting and cell confluence using the S3 Incucyte Live-Cell Analysis System (Sartorius AG, Göttingen, Germany). Cell lines were seeded in triplicates into 96-well plates, and confluence and mKATE2-NLS numbers were determined for 5 days. Pearson correlation (r) was calculated using Prism software (Boston, MA, USA).

**Figure 2 ijms-25-09275-f002:**
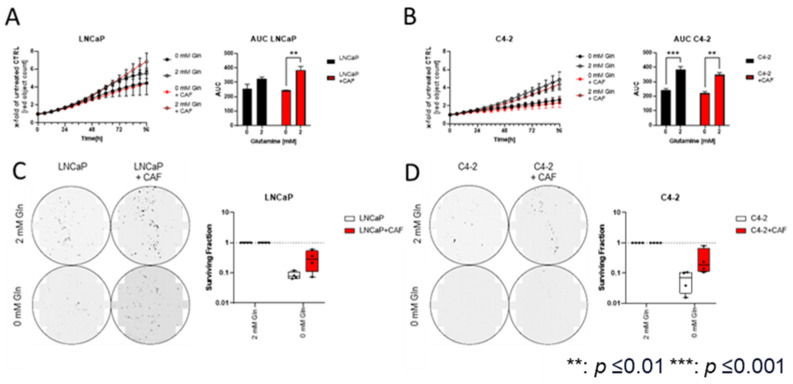
Influence of glutamine (Gln) deprivation on selected prostate cancer (PCa) cell growth co-cultured with cancer-associated fibroblast (CAF) cells. (**A**) Influence of Gln deprivation on the LNCaP cell proliferation in the presence or absence of CAF cells. Red object count assessed proliferation for 96 h. Relative changes in AUC values were calculated from the growth curve experiments. Data were plotted as mean ± SEM of three biological replicates. Significant differences were identified using two-way ANOVA. *p*-values ≤ 0.05 were considered significant. **: *p* ≤ 0.01. (**B**) Influence of Gln deprivation on the C4-2 cell proliferation in the presence or absence of CAF cells. Red object count assessed proliferation for 96 h. Relative changes in AUC values were calculated from the growth curve experiments. Data were plotted as mean ± SEM of three biological replicates. Significant differences were identified using two-way ANOVA. *p*-values ≤ 0.05 were considered significant. **: *p* ≤ 0.01, ***: *p* ≤ 0.001 (**C**) Representative images of the clonogenic assays of the cell lines LNCaP and colony-forming efficiency (CFE) calculated from the clonogenic assays of LNCaP cells co-cultured with CAF cells after Gln deprivation. Colony number (≥50 cells/colony) was scored 10 days after plating. The results are expressed as box and whisker plots (min to max) of 4 biological replicates and are compared with monocultured cells. (**D**) Representative images of the clonogenic assays of the cell lines C4-2 cells and colony-forming efficiency (CFE) calculated from the clonogenic assays of C4-2 cells co-cultured with CAF cells after Gln deprivation. Colony number (≥50 cells/colony) was scored 10 days after plating. The results are expressed as box and whisker plots (min to max) of 4 biological replicates and are compared with monocultured cells.

**Figure 3 ijms-25-09275-f003:**
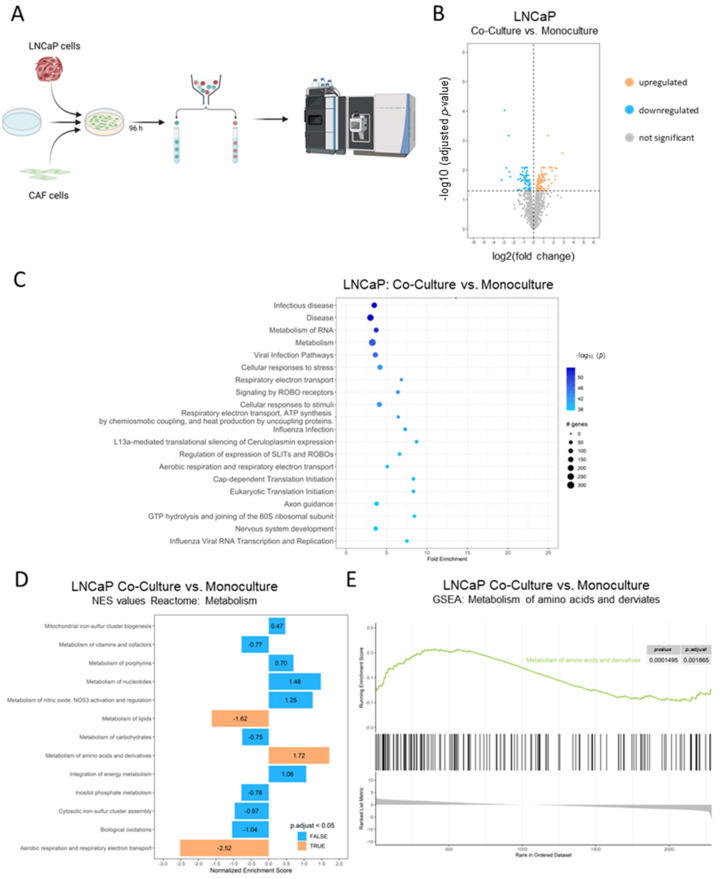
PathfindR and GSEA analysis results for differentially expressed genes after co-culturing LNCaP and CAF cells. (**A**) Graphical representation of the experimental procedure. Created with BioRender.com (accessed on 26 August 2023). LNCaP mKATE2-nls cells (1 × 10^6^) and CAF eGFP-nls cells (5 × 10^5^) were seeded into a 10 cm^2^ dish and cultured for 96 h. The different cells were then sorted and monocultured and the sorted cells were lysed for mass spectrometry. (**B**) Volcano plot of differentially expressed proteins in LNCaP cells after co-culturing the cells for 96 h with CAFs. Colour coding: gray = no statistically significant difference and not differentially expressed; blue = statistically significantly downregulated proteins, red = statistically significantly upregulated proteins. (**C**) The top 20 enriched pathways were identified by “pathfindR” pathway analysis using Reactome pathways, ordered by −log10(Padj) in LNCaP cells after co-culturing the cells for 96 h with CAF cells. (**D**) Selected metabolic gene sets enriched in co-cultured LNCaP cells using the REACTOME database. (**E**) Selected GSEA plots of metabolism of amino acids and derivates for LNCaP cells co-cultured with CAF.

**Figure 4 ijms-25-09275-f004:**
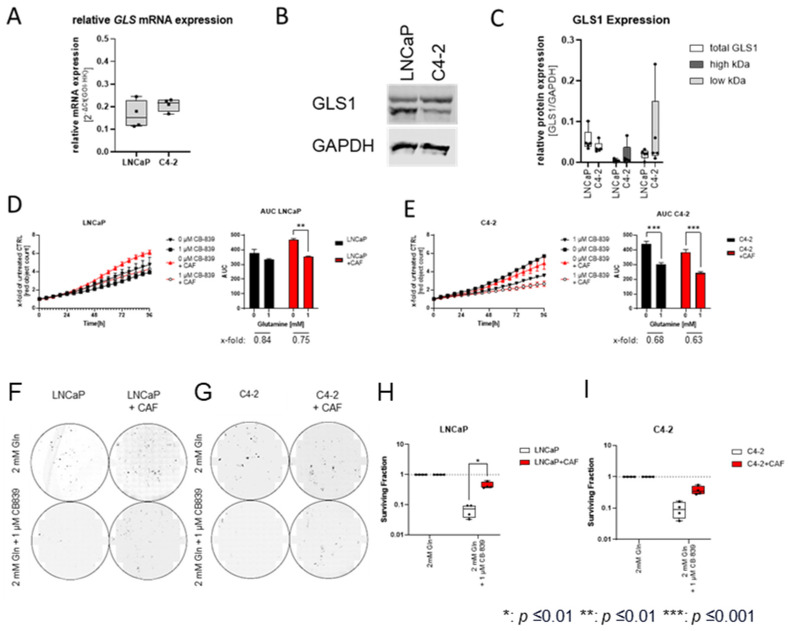
Influence of CB839 on selected prostate cancer (PCa) cell proliferation co-growth with cancer-associated fibroblast (CAF) cells. (**A**) qPCR analysis of relative GLS mRNA levels in LNCaP and C4-2 cells (*n* = 4). mRNA levels were normalised to HPRT1. The results are expressed as box and whisker plots (min to max) of the 4 biological replicates. (**B**) Representative Western blots of GLS1 (55–65 kDa) and the housekeeper GAPDH (37 kDa). Chameleon duo pre-stained protein ladder (Ladder) was used as protein size standard. (**C**) Densiometric Western blot analysis of GLS1 protein levels in LNCaP and C4-2 cells (*n* = 4). Protein levels were normalised to GAPDH. The results are expressed as box and whisker plots (min to max) of 4 biological replicates. (**D**) Influence of CB-839 on the LNCaP cell proliferation in the presence or absence of CAF cells. Red object count assessed proliferation for 96 h. Relative changes in AUC values were calculated from the growth curve experiments. Data were plotted as mean ± SEM of three biological replicates. Significant differences were identified using two-way ANOVA. *p*-values ≤ 0.05 were considered significant. **: *p* ≤ 0.01. (**E**) Influence of CB-839 on the C4-2 cell proliferation in the presence or absence of CAF cells. Red object count assessed proliferation for 96 h. Relative changes in AUC values were calculated from the growth curve experiments. Data were plotted as mean ± SEM of three biological replicates. Significant differences were identified using two-way ANOVA. *p*-values ≤ 0.05 were considered significant. ***: *p* ≤ 0.001. (F + G) Representative images of the clonogenic assays of the cell lines LNCaP (**F**) and C4-2 (**G**). (H + I) Colony-forming efficiency (CFE) calculated from the clonogenic assays of LNCaP (**H**) and C4-2. (**I**) cells co-cultured CAF cells after CB-839 treatment. Colony number (≥50 cells/colony) was scored 10 days after plating. The results are expressed as box and whisker plots (min to max) of 4 biological replicates and are compared with monocultured cells *: *p* ≤ 0.01.

**Figure 5 ijms-25-09275-f005:**
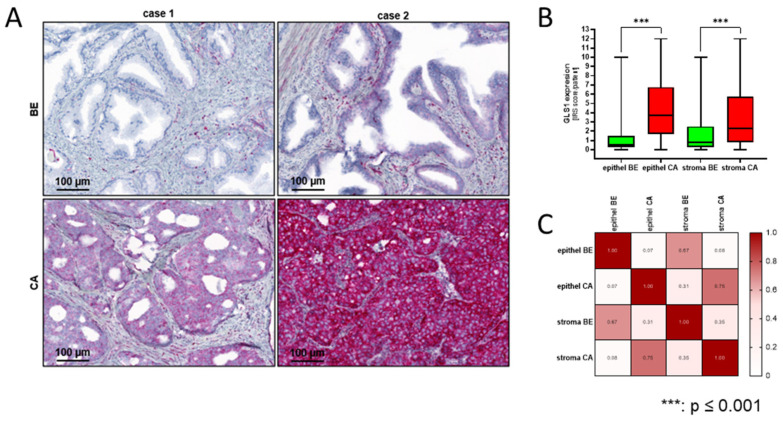
GLS1 expression is elevated in malignant epithelial and stroma prostate areas. (**A**) Immunohistochemical staining for GLS1 of representative benign and malignant prostate tissue. Scale bar = 100 μm. (**B**) Quantification of GLS1 after immunohistochemistry (IHC) staining of benign and malignant prostate tissue. Staining was evaluated using the immunoreactivity score (IRS), ranging from 0 to 12 for GLS1. The results are expressed as box and whisker plots (min to max). Significant differences were identified using one-way ANOVA. *p*-values ≤ 0.05 were considered significant. ***: *p* < 0.001. (**C**) Pearson correlation of GLS1 expression in benign and malignant PCa areas. The r-values are displayed in a heat map.

**Figure 6 ijms-25-09275-f006:**
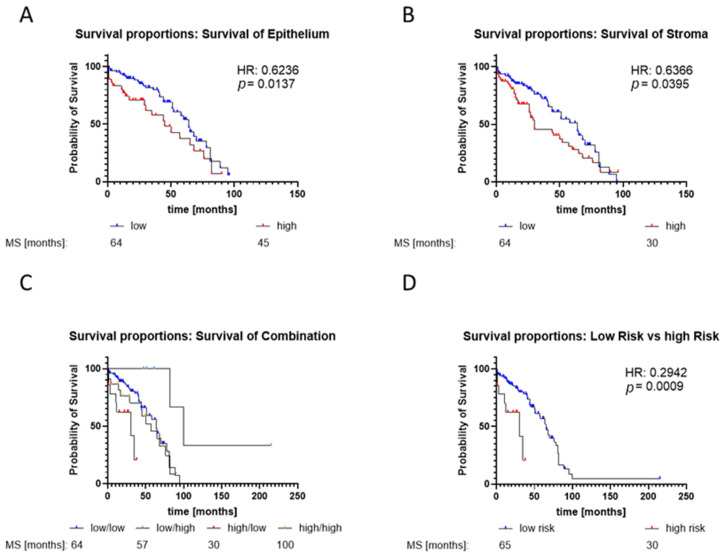
Kaplan–Meier analysis of the prostate cancer (PCa) cohort according to the GLS1 expression reveals a high-risk group with bad clinical outcomes. Median overall survival (MS) analysis was performed based on GLS1 expression in the epithelium (**A**), stroma (**B**), and combined expression in the epithelium and stroma (**C**). For combination analysis, data were grouped into the following expression patterns (epithelium/stroma): high/high, high/low, low/high, and low/low. The median GLS1-IRS score was selected as the threshold. (**D**) Kaplan–Meier analysis of the identified high-risk group (high/low) compared with the rest of the PCa cohort. Abbreviations: HR—hazard ratio (log-rank). MS—median overall survival.

**Table 1 ijms-25-09275-t001:** Baseline characteristics of PCa patient cohort.

	All	HSPC	CRPC
Patient Number	108	26	82
Median age at primary diagnosis, years	71	72	71
Median PSA at primary diagnosis, ng/mL (Interquartile range IQR)	17	6.3	32.5
	(6.8; 73.0)	(2.7; 10.2)	(8.9; 101.6)
Neuroendocrine differentiation at primary diagnosis, %	1	0	1
Presence of bone metastases at primary diagnosis, %	25	3.8	32
Presence of lymph node metastases at primary diagnosis, %	14	12	11.7
Presence of organ metastases at primary diagnosis, %	1	1	1
Median overall survival since the start of primary therapy, months	59	47	81

## Data Availability

The data presented in this study are openly available in PCaSilentPartners at https://doi.org/10.5281/zenodo.13208429, reference number 7c76bd2.

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
