# Peer review of "Prostate Cancer’s Silent Partners: Fibroblasts and Their Influence on Glutamine Metabolism Manipulation"

_ijms, 2024, doi:10.3390/ijms25179275_

Round 1

Reviewer 1 Report

Comments and Suggestions for Authors

Topic of presented manuscript is interesting and its result has clinical potentials. But some point should be taken for the improvement its quality.

GLS1 role should be significantly more described in the introduction

Figure1: can please show values of Person correlation coefficients? In the text SI figures are not referenced and discussed. Line 496 both mentioned figures are labelled as S1.

Difference between LNCaP and C4-2 cells should be significantly more discussed, for example by using bellow works

Xie BX, Zhang H, Yu L, Wang J, Pang B, Wu RQ, Qian XL, Li SH, Shi QG, Wang LL, Zhou JG. The radiation response of androgen-refractory prostate cancer cell line C4-2 derived from androgen-sensitive cell line LNCaP. Asian J Androl. 2010 May;12(3):405-14. doi: 10.1038/aja.2009.91. Epub 2010 Feb 2. PMID: 20118949; PMCID: PMC3739257.

term “multicolour PCa-fibroblast co-culture/model“ should be better explained.

Experimental section

293T cells are mentioned in the experimental section, but no in the others part of manuscript.

CAF should be labelled as hTERT PF179T CAF not hTERT PF179T C.A.F. The

Minor some abbreviations (GLS1 and OXHOS) are not defined

Line 92 mKATE2 fluorescent protein, position of excitation and emission maximum should be mentioned.

Author Response

On behalf of all the authors, we would like to take this opportunity to express our sincere gratitude to the reviewers who identified areas of our manuscript that needed correction or modification. Their insightful comments have led to an improvement in our manuscript. Below, you can find the detailed response to their comments:

  1. GLS1 role should be significantly more described in the introduction
  • As suggested, we added more information about GLS1 into the introduction.
  1. Figure1: can please show values of Person correlation coefficients? In the text SI figures are not referenced and discussed. Line 496 both mentioned figures are labelled as S1.
  • In response to the reviewer's comments, we have made several adjustments to improve the clarity and readability of Figure 1D. Specifically, we increased the font size of the Pearson correlation coefficients and the p-values to enhance their visibility. Additionally, we have incorporated these values directly into the text for clearer presentation and reference.
  • Regarding the abbreviation "SI figures" mentioned in the comment, we were initially unsure of its meaning. Upon reflection, we surmise that the reviewer referred to the significance indicators typically denoted by asterisks in graphical representations. We have included these asterisks in the illustration to signify the statistical significance levels.
  • The labelling issue has been corrected.
  1. The difference between LNCaP and C4-2 cells should be significantly more discussed, for example by using bellow works

Xie BX, Zhang H, Yu L, Wang J, Pang B, Wu RQ, Qian XL, Li SH, Shi QG, Wang LL, Zhou JG. The radiation response of androgen-refractory prostate cancer cell line C4-2 derived from androgen-sensitive cell line LNCaP. Asian J Androl. 2010 May;12(3):405-14. doi: 10.1038/aja.2009.91. Epub 2010 Feb 2. PMID: 20118949; PMCID: PMC3739257.

  • We thank the reviewer for their suggestion and added more discussion to the manuscript using the suggested work.
  1. term "multicolour PCa-fibroblast co-culture/model "should be better explained.
  • We removed the multicolour and added a more detailed explanation to the manuscript.
  1. 293T cells are mentioned in the experimental section, but no in the others part of manuscript.
  • 293T cells were used for lentiviral production. So, they are mentioned in the lentiviral production section. As they are essential to virus production, the author thinks mentioning them in the used cell lines section is necessary.
  1. CAF should be labelled as hTERT PF179T CAF not hTERT PF179T C.A.F. The
  • We changed the spelling as suggested
  1. Minor some abbreviations (GLS1 and OXHOS) are not defined
  • Abbreviations have been added
  1. Line 92 mKATE2 fluorescent protein, position of excitation and emission maximum should be mentioned.
  • The excitation and emission max had been added.

Reviewer 2 Report

Comments and Suggestions for Authors

The presented study covers a significant gap by exploring the interaction between CAFs and glutamine metabolism in prostate cancer. In this regard, the authors have made a significant set of highly advanced assessments including proteomic analysis, immunohistochemical, flow cytometric, PCR, WB, and clonogenic analysis. The applicability of multicolor PCa-fibroblast co-culture to assess the influence of CAFs on PCa growth is innovative and further inclusion of hormone-sensitive and castration-resistant PCa cell lines (LNCaP and C4-2) adds depth to the study. The presented findings have important implications for cancer biology and the development of targeted therapies. The reviewer has some concerns that could help in improving the clarity and rigor of this study:

-          In the introduction part, the authors should highlight the interaction between CAFs and glutamine metabolism in other types of cancers to provide a broader context.

-          The authors should clarify the conditions under which Gln metabolism was manipulated …how were Gln deprivation and GLS1 inhibition specifically implemented, and what controls were used? particularly the specific concentrations and durations used for Gln deprivation and GLS1 inhibition

-          the significance of the differences between cell lines and treatment conditions should be associated and clarified… how the multicolor system differentiates between PCa cells and CAFs during analysis

-          it would be beneficial ro show the response of dose-response GLS1 inhibition

-          regarding the patient tissues, it would be beneficial to provide and discuss the variability between samples.

-          differential impact of CAFs on LNCaP and C4-2 cells raises concerns about the mechanism involved… do the authors consider this attribute to variations in Gln metabolism between hormone-sensitive and castration-resistant cells or other factors e.g., differential expression of CAF-derived factors.

-          Why the authors have not proceeded with metabolomic analyss and connected with the Gln metabolism pathways? This could identify key metabolites that are altered, providing insight into the metabolic rewiring induced by CAFs

-          Have the authors tried to check the role of CAF-secreted factors (e.g., cytokines, growth factors) in modulating Gln metabolism? E.g., conditioned media from CAFs could be used to treat PCa cells, followed by analysis of Gln uptake and utilization.

-          The authors should extend their discussion to include the limitations of these findings.. the challenge to explore in animal models? Also, the authors should also consider the role of other metabolic pathways identified by proteomic analysis and their influence on stromal cells in TME? The authors should also discuss how glutamine metabolism is implicated in other cancers  based on their findings.

-          Also, it would be beneficial to shortly associate the mechanisms that might be activated in response to Gln deprivation or GLS1 inhibition e.g., cancer cells might shift to alternative nutrient sources or metabolic pathways (e.g., glucose or fatty acid metabolism), which could limit the efficacy of GLS1-targeted therapies.

-          Finally, the article is missing a comprehensive conclusion part… the authors should inlcud the future research directions, including potential in vivo studies or clinical trials to validate GLS1 as a therapeutic target.

Comments on the Quality of English Language

Minor editing of English language required.

Author Response

On behalf of all the authors, we would like to take this opportunity to express our sincere gratitude to the reviewers who identified areas of our manuscript that needed correction or modification. Their insightful comments have led to an improvement in our manuscript. Below, you can find the detailed response to their comments:

  1. -          In the introduction part, the authors should highlight the interaction between CAFs and glutamine metabolism in other types of cancers to provide a broader context.
  • As suggested, we provided more information about the interaction between CAFs and glutamine metabolism in different cancer entities
  1. -          The authors should clarify the conditions under which Gln metabolism was manipulated …how were Gln deprivation and GLS1 inhibition specifically implemented, and what controls were used? particularly the specific concentrations and durations used for Gln deprivation and GLS1 inhibition
  • -Detailed information about the treatment has been added to the material and methods section 4.3
  1. -          the significance of the differences between cell lines and treatment conditions should be associated and clarified… how the multicolor system differentiates between PCa cells and CAFs during analysis
  • The cell lines have been labelled with two fluorescent proteins (FP fused to a nuclear localisation signal). Therefore, the nuclei are coloured, and the IncuCyte® S3 Live-Cell Analysis System can detect and count the nuclei of the cells. This information is in the manuscript in 4.4. In addition, we added more information about our two-colour system to the manuscript. In order to guarantee the same test conditions, the tests were carried out in RPMI1640 medium. Previous publications have shown that C4-2 also survives in RPMI medium and does not affect growth behaviour (PMID: 26499105, PMID: 14584899, PMID: 34193425). The test procedure is described in 4.3.
  1. -          it would be beneficial ro show the response of dose-response GLS1 inhibition
  • As suggested, dose-response curves have been added to Figures S2E
  1. -          regarding the patient tissues, it would be beneficial to provide and discuss the variability between samples.
  • Baseline characteristics of the PCa-patients cohort have been added to Table 1.
  1. -          differential impact of CAFs on LNCaP and C4-2 cells raises concerns about the mechanism involved… do the authors consider this attribute to variations in Gln metabolism between hormone-sensitive and castration-resistant cells or other factors e.g., differential expression of CAF-derived factors.
  • The underlying mechanism appears complex, involving CAF-derived factors and cell-cell interactions that increase proliferation in LNCaP cells, raising energy demands. Further studies are required to elucidate this mechanism fully. We have discussed these points in our manuscript and outlined potential directions for future research.
  1. -          Why the authors have not proceeded with metabolomic analyss and connected with the Gln metabolism pathways? This could identify key metabolites that are altered, providing insight into the metabolic rewiring induced by CAFs
  • We appreciate the suggestion and believe that incorporating multiple omic analyses, including metabolomics in follow-up studies, would complement our current findings and enhance the understanding of the role of CAFs in prostate cancer metabolism. We added this intention into our outlook in the limitation part.
  1. -          Have the authors tried to check the role of CAF-secreted factors (e.g., cytokines, growth factors) in modulating Gln metabolism? E.g., conditioned media from CAFs could be used to treat PCa cells, followed by analysis of Gln uptake and utilisation.
  • We opted not to use a conditioned media experimental design because we aimed to preserve cell-cell contact, a critical aspect of epithelial cell communication with their microenvironment. Relying solely on supernatants or secreted factors would have overlooked this vital interaction. Furthermore, it is well-documented that cytokines and growth factors can modulate cellular metabolism, as highlighted in several studies (reviewed in PMID: 30249998). Therefore, maintaining direct cellular interactions was essential to reflect the complex communication mechanisms accurately.
  1. -          The authors should extend their discussion to include the limitations of these findings.. the challenge to explore in animal models? Also, the authors should also consider the role of other metabolic pathways identified by proteomic analysis and their influence on stromal cells in TME? The authors should also discuss how glutamine metabolism is implicated in other cancers  based on their findings.
  • As suggested, we have discussed the study's limitations and challenges in exploring these findings in animal models. Our proteomic analysis revealed a significant enrichment, specifically in the "Metabolism of amino acids and derivatives" pathway in LNCaP cells following co-culture, which led us to focus on this metabolic pathway exclusively without considering others. In contrast, our results indicated that fibroblast metabolism remained unaltered (Figure S2B). Additionally, we have incorporated information on glutamine metabolism observed in other cancers into the discussion.
  1. -          Also, it would be beneficial to shortly associate the mechanisms that might be activated in response to Gln deprivation or GLS1 inhibition e.g., cancer cells might shift to alternative nutrient sources or metabolic pathways (e.g., glucose or fatty acid metabolism), which could limit the efficacy of GLS1-targeted therapies.
  • A paragraph about a possible metabolic shift has been added to the discussion.
  1. -          Finally, the article is missing a comprehensive conclusion part… the authors should inlcud the future research directions, including potential in vivo studies or clinical trials to validate GLS1 as a therapeutic target.
  • Possible future directions have been added to the discussion.

Reviewer 3 Report

Comments and Suggestions for Authors

The manuscript entitled “Prostate Cancer's Silent Partners: Fibroblasts and Their Influence on Glutamine Metabolism Manipulation” focuses on the importance of interaction between PCa and CAFs in tumor microenvironment for potential application of therapeutics targeting Gln metabolism. The topic is within the scope of the journal, highlights the necessity of evaluating the therapeutic potential of metabolic modifiers in systems that simulate the histological context relevant for PCa and would be interesting to clinicians and researchers engaged in PCa management and research.  

The manuscript is well structured, well written and fluently presented. There are merely some minor corrections that should be made and some clarifications are required in order to make the manuscript more informative (in order of appearance):

- The authors should pay attention to the usage of abbreviations; some are not commonly used and widely known and should be defined at first usage: CAC (line 57), OXPHOS (line 72), etc. Additionally, the second C in CAC is for “cycle”, so “CAC cycle” should be corrected to “CAC”. In line 71, “C.A.F.s” should be corrected to “CAFs” in order for abbreviations to be uniform throughout the manuscript.

- There is an error in Figure 2. The same image is used for Figure 2A and 2B (the left part of both).

- GLS1 should be defined and its role should be explained in the Introduction section.

- There is another important interpretation of Figure 4C, apart from “comparable expression of GLS1 protein”: there is obviously a difference in the ration of GLS1 isoforms which is crucial for therapeutic resistance and PCa progression. The Results should include a comment on this finding and it should be discussed in the context of other findings as well.

- Line 476: the abbreviation “AD” should be defined.

- Lines 482-484: If only HPRT1 was used as an internal reference, why was Taqman assay for TBP included in qPCR analysis?

Author Response

On behalf of all the authors, we would like to take this opportunity to express our sincere gratitude to the reviewers who identified areas of our manuscript that needed correction or modification. Their insightful comments have led to an improvement in our manuscript. Below, you can find the detailed response to their comments:

  1. - The authors should pay attention to the usage of abbreviations; some are not commonly used and widely known and should be defined at first usage: CAC (line 57), OXPHOS (line 72), etc. Additionally, the second C in CAC is for "cycle", so "CAC cycle" should be corrected to "CAC". In line 71, "CAFs" should be corrected to "CAFs" in order for abbreviations to be uniform throughout the manuscript.
  • We want to thank the reviewer for his comment. We corrected the abbreviations.
  1. - There is an error in Figure 2. The same image is used for Figure 2A and 2B (the left part of both).
  • We want to thank the reviewer for pointing out this mistake and apologise. We corrected the figure.
  1. - GLS1 should be defined and its role should be explained in the Introduction section.
  • As suggested, we added more information about GLS1 to the introduction.
  1. - There is another important interpretation of Figure 4C, apart from "comparable expression of GLS1 protein": there is obviously a difference in the ration of GLS1 isoforms which is crucial for therapeutic resistance and PCa progression. The Results should include a comment on this finding and it should be discussed in the context of other findings as well.
  • We agree with the reviewer that different isoforms play an important role in PCa. Unfortunately, based on the Western blot, we cannot be sure whether the 2 bands are GLS KGA and GAC, nor can we say what GLS forms they are by Western blot. We have analysed the western blots, and unfortunately, we cannot calculate a significant change in the unbanned GLS forms. We could confirm that three different siRNAs against GLS1 can be used to know-down both proteins (Beier et al. 2024). However, we added a short discussion about both KGA and GAC to the discussion section.

  1. - Line 476: the abbreviation "AD" should be defined.
  • We replaced AD with "RNAse and DNase-free distilled water.
  1. - Lines 482-484: If only HPRT1 was used as an internal reference, why was Taqman assay for TBP included in qPCR analysis?
  • We want to thank the reviewer for pointing out this mistake and would like to apologise for the mistake. This issue was a copy-paste error, and we removed the primer set.

Round 2

Reviewer 1 Report

Comments and Suggestions for Authors

Minor

Line 171, 172 and 179 S_Figure 1 should be called as Figure S1

Author Response

On behalf of all the authors, we would like to take this opportunity to express our sincere gratitude to the reviewers who identified areas of our manuscript that needed correction or modification. Their insightful comments have led to an improvement in our manuscript. Below, you can find the detailed response to their comments:

  • Line 171, 172 and 179 S_Figure 1 should be called as Figure S1

We would like to thank the reviewer for bringing these errors to our attention. The errors have been corrected.

Reviewer 2 Report

Comments and Suggestions for Authors

the authors have adequately addressed the concerns raised by the reviewer and the manuscript has been improved.

Author Response

On behalf of all the authors, we would like to take this opportunity to express our sincere gratitude to the reviewers who identified areas of our manuscript that needed correction or modification. Their insightful comments have led to an improvement in our manuscript. Below, you can find the detailed response to their comments: